# Characterization and Probiotic Potential of *Levilactobacillus brevis* DPL5: A Novel Strain Isolated from Human Breast Milk with Antimicrobial Properties Against Biofilm-Forming *Staphylococcus aureus*

**DOI:** 10.3390/microorganisms13010160

**Published:** 2025-01-14

**Authors:** Ivan Iliev, Galina Yahubyan, Elena Apostolova-Kuzova, Mariyana Gozmanova, Daniela Mollova, Iliya Iliev, Lena Ilieva, Mariana Marhova, Velizar Gochev, Vesselin Baev

**Affiliations:** 1Department of Biochemistry and Microbiology, Faculty of Biology, University of Plovdiv, Tzar Assen 24, 4000 Plovdiv, Bulgaria; 2Department of Molecular Biology, Faculty of Biology, University of Plovdiv, Tzar Assen 24, 4000 Plovdiv, Bulgaria

**Keywords:** *Levilactobacillus brevis*, *Staphylococcus aureus*, LAB, genome sequencing, probiotic and anti-biofilm capacity

## Abstract

Lactobacillus is a key genus of probiotics commonly utilized for the treatment of oral infections The primary aim of our research was to investigate the probiotic potential of the newly isolated *Levilactobacillus brevis* DPL5 strain from human breast milk, focusing on its ability to combat biofilm-forming pathogens such as *Staphylococcus aureus*. Employing in vitro approaches, we demonstrate *L. brevis* DPL5′s ability to endure at pH 3 with survival rates above 30%, and withstand the osmotic stress often found during industrial processes like fermentation and freeze drying, retaining over 90% viability. The lyophilized cell-free supernatant of *L. brevis* DPL5 had a significant antagonistic effect against biofilm-producing nasal strains of *Staphylococcus aureus*, and it completely eradicated biofilms at subinhibitory concentrations of 20 mg·mL^−1^. Higher concentrations of 69 mg·mL^−1^ were found to have a 99% bactericidal effect, based on the conducted probability analysis, indicating the production of bactericidal bioactive extracellular compounds capable of disrupting the biofilm formation of pathogens like *S. aureus*. Furthermore, genome-wide sequencing and analysis of *L. brevis* DPL5 with cutting-edge Nanopore technology has uncovered over 50 genes linked to probiotic activity, supporting its ability to adapt and thrive in the harsh gut environment. The genome also contains multiple biosynthetic gene clusters such as lanthipeptide class IV, Type III polyketide synthase (T3PKS), and ribosomally synthesized, and post-translationally modified peptides (RiPP-like compounds), all of which are associated with antibacterial properties. Our study paves the way for the further exploration of DPL5, setting the stage for innovative, nature-inspired solutions to combat stubborn bacterial infections.

## 1. Introduction

Human breast milk is a multifaceted and adaptive substance that represents the pinnacle of infant feeding. It offers the ideal nutrient content, immunological elements, and bioactive constituents essential for the growth, development, and safeguarding of the newborn [1,2]. There is increasing emphasis on the importance of human breast milk as a source of beneficial bacteria that contribute to a healthy gut microbiome in babies [3]. We acknowledge it as an excellent source of probiotics because the bacteria it contains fulfill essential criteria for probiotics: human origin, adaptability to milk substrates, and a documented history of safe, extended consumption by newborns [4,5]. Numerous bacterial families, such as Streptococcaceae, Pseudomonadaceae, Staphylococcaceae, Lactobacillaceae, and Oxalobacteraceae, are prevalent in human breast milk [6,7,8]. *Lactobacillus and Bifidobacterium*, notable genera among the several bacteria species present in human breast milk, are recognized for their probiotic potential [9,10,11]. Research indicates that breast milk serves as a natural reservoir for multiple species of *Lactobacillus*, including *L. rhamnosus*, *L. plantarum*, *L. fermentum*, *L. gasseri*, *L. salivarius*, *L. reuteri*, *L. casei*, *and L*. *brevis* [9,10,12,13].

Lactic acid bacteria (LAB) are Gram-positive, catalase-negative, non-spore forming, coccus- or rod-shaped bacteria that produce organic acids after glucose fermentation [11,14,15]. Although most probiotics belong to the LAB family, not all group members possess probiotic properties. As established by the World Health Organization (WHO), probiotics are “live microorganisms which when administered in adequate amounts confer a health benefit on the host” [16]. For an LAB to be acknowledged as a probiotic, it must survive the harsh conditions of the gastrointestinal tract. Probiotic bacteria can produce antimicrobial substances, adhere to intestinal cells, and inhibit the adhesion of pathogenic bacteria, thereby ensuring their survival [17,18] Numerous Lactobacillus strains produced from human breast milk have been employed as probiotics, exhibiting effectiveness in preventing and treating infections, inflammation, and allergies [19]. Due to their probiotic properties, several strains have been generally recognized as safe by the Food and Drug Administration and qualified presumption of safety by the European Food Safety Authority [20].

Many studies have demonstrated that Lactobacillus species may produce lactic acid, hydrogen peroxide, and bacteriocins, among other antimicrobial compounds, which can hinder harmful bacterial strains’ growth and biofilm formation. Bacteriocins are antimicrobial peptides produced by bacteria, including Lactobacillus, and display bacteriostatic or bactericidal activity against other bacteria [21]. Ribosomes manufacture bacteriocins. These peptides disrupt the cell membrane of target bacteria, leading to cell death. Studies have demonstrated that Lactobacillus-derived bacteriocins and other mechanisms of action, such as competition for adhesion sites and nutrients, act as effective inhibitors of growth and biofilm formation in *Staphylococcus aureus* [22,23].

*S. aureus* is a Gram-positive bacterium that is widespread in the environment and frequently colonizes humans’ skin and mucous membranes. It is a major human pathogen capable of causing many infections, ranging from mild skin infections to life-threatening conditions such as pneumonia, endocarditis, and sepsis [24,25]. The ability of *S. aureus* to form biofilms, complex microbial communities attached to surfaces, contributes to its virulence and antibiotic resistance. Biofilms provide a protective barrier for bacteria, making them more difficult to eradicate with conventional antibiotics [26,27].

Despite the limited research on the probiotic potential of *L. brevis* from human breast milk, existing studies suggest that using Lactobacillus and its bacteriocins as natural antimicrobial agents to control *S. aureus* infections is promising, and it could help cut down on antibiotic use and the development of antibiotic resistance [28]. The study aims to look at the genomic structure and probiotic potential of *L. brevis* from human breast milk and evaluate its antibacterial properties against *S. aureus*. Specifically, this research seeks to evaluate the antimicrobial activity of *L. brevis*-derived cell-free supernatant and its ability to prevent biofilm formation. By exploring the interaction between *Lactobacillus* metabolites and *S. aureus*, this study aims to contribute to the growing body of knowledge surrounding the use of probiotics and their metabolites in managing bacterial infections, particularly those involving biofilm formation.

## 2. Materials and Methods

### 2.1. Breast Milk Sample Collection

Breast milk samples were collected from nursing mothers within the first six months postpartum. All participants gave informed consent for their inclusion. The research was conducted by the Declaration of Helsinki, and the study protocol was approved by the Ethics Committee at the University of Plovdiv (Approval No. 6/06.10.2022). The participants were healthy women who had delivered full-term infants either via vaginal birth or cesarean section. The mothers were asked to wash their breasts with water to collect the milk samples and then express 15–20 mL of milk into a sterile container. These containers were stored at 4 °C until retrieved and transported to the laboratory. All milk samples were processed within 24 h of collection.

### 2.2. Bacterial Isolation and Identification and Culture Conditions

Standard laboratory protocols were followed to collect and identify LAB [2]. At first, samples with 1 mL volume were enriched by 24 h anaerobic incubation in 9 mL De Man Rogosa and Sharpe (MRS) liquid medium. Tenfold serial dilutions were spread plated on De Man Rogosa and Sharpe (MRS) agar and incubated anaerobically at 37 °C for 72 h in a 2.5 L anaerobic jar with Anaerocult A reagent (Merck, Darmstadt, Germany). Single colonies that developed on the MRS agar were purified by streaking on new dishes with MRS agar [2]. The preliminary identification of *Lactobacillus* strains included Gram staining and test for catalase production. According to Bergey’s Manual of Systematic Bacteriology, bacterial isolates were initially identified based on their colony morphology, cultural characteristics, and microscopic appearance [29]. Those determined to be Gram-positive, non-spore-forming, and catalase-negative rods were presumptively identified as *Lactobacillus* spp. The isolated strains were preserved as stock cultures at −20 °C in MRS broth (Merck, Darmstadt, Germany) with 15% glycerol (Merck, Darmstadt, Germany) for further analysis. Species identity was confirmed by GeN III plate with Biolog’s OmniLog ID System (Swampscott, MA, USA) instrument [30]. Isolate DPL5, identified as *Lactobacillus brevis*, was chosen for further analysis.

*Staphylococcus aureus* strains used for the inhibitory tests were obtained from the Microorganisms Culture Collection of the Department of Biochemistry and Microbiology (Paisii Hilendarski Plovdiv University, Plovdiv, Bulgaria). A total of 33% were MRSA strains to detect differences in the effects of *L. brevis* on cell proliferation and biofilm formation. Information regarding the antimicrobial susceptibility of the isolates is presented in Appendix A. The strains were cultured in tryptic soy broth (TSB) supplemented with 1% (*w*/*v*) glucose and 2% (*w*/*v*) NaCl [31]. *S. aureus* inoculums were standardized by homogenization in sterile 0.85% NaCl, and all suspensions were diluted to 0.5 × 10^8^ CFU mL^−1^ using McFarland densitometer DEN-1 (Grant-bio, Cambridge, England) prior to inoculation [24].

### 2.3. Probiotic Potential Assessment

#### 2.3.1. Assimilation of Different Types of Carbon Sources and Osmotic Sensitivity

The metabolic profile of *L. brevis* DPL5 was analyzed using Biolog’s Phenotype MicroArrays™ (PM) in conjunction with the Omnilog™ (Orlando, FL, USA) instrument. During the experiment, the samples were incubated and read continuously at 20-min intervals for 24 h in the Omnilog instrument. Data collection involved a two-step approach. First, Omnilog was used to measure the OD in each well and directly assess cell proliferation. A redox-sensitive dye was used to measure the color change in each well based on NADH production, allowing the determination of metabolic activity. The simultaneous measurements of both cell density and color change in the dye make it possible to understand the differences between metabolism and cell growth in terms of their phenotypic differences. A Gen III plate with inoculation fluid A was used in the experiments.

#### 2.3.2. Bile Salt Tolerance

The strain’s capacity to endure various amounts of bile salts was estimated at three distinct concentrations of bile salts: 0.3%, 1%, and 3%. MRS with bile salts was prepared and inoculated with overnight bacterial suspension (dilution factor 10). Optical density was assessed 24 h post-inoculation at a wavelength of 600 nm using the Beckman Coulter DU 730 spectrophotometer (Brea, CA, USA), and bacterial growth levels were documented [31]. The assay was performed twice each in triplicate.

#### 2.3.3. Acid Tolerance

The capacity to endure acidic conditions was evaluated by exposing samples to pH 3, 5, 7, 9 and 10 in MRS broth inoculated with overnight bacterial suspension (dilution factor 10). Optical density was assessed 24 h post-inoculation at a wavelength of 600 nm with the Beckman Coulter DU 730 spectrophotometer (Brea, CA, USA). The readings were recorded at 24 h of incubation, and this process was repeated twice for each sample.

#### 2.3.4. Determination of Antibiotic Susceptibility

The method described by Sharma and collaborators [32] was used to evaluate the antibiotic susceptibility of the *L. brevis* DPL5. The susceptibility to ten commonly used clinical antibiotics (Erythromycin, Amikacin, Gentamicin, Kanamycin, Amoxicillin, Ampicillin, Penicillin, Ciprofloxacin, Chloramphenicol, and Ceftriaxone) was determined using the disc diffusion method. Antibiotics were obtained from Oxoid (Thermo Fisher Scientific, Wesel, Germany). Active cultures were prepared by adjusting the suspension densities to McFarland 0.5. Then, 100 µL of the isolate was spread onto MRS agar (Merck, Darmstadt, Germany). Subsequently, three antibiotic discs were placed on inoculated MRS agar and incubated at 37 °C for 24 h. After incubation, the diameter of the zone of inhibition around each disc was measured. The zone of inhibition represents the area where the growth of the microorganism was restrained, following the guidelines of the Clinical and Laboratory Standards Institute [33]. The multiple antibiotic resistance (MAR) index was calculated using the method of Gyorgy et al. [34].

### 2.4. Preparation and Lyophilization of L. brevis Cell-Free Supernatant (LCFS)

*Lactobacillus brevis* DPL5 was cultured in 200 mL MRS broth for 48 h. The cells were separated by centrifugation at 10,000× *g* for 20 min at 4 °C. The supernatant was filtered through a 0.22 µm nitrocellulose membrane. To eliminate the effect of lactic acid, the pH of the supernatant was neutralized [35]. The sterile supernatant and pure MRS medium were frozen at −80 °C and lyophilized (Labcocno FreeZone 4.5; Labconco Corporation, Kansas City, MI, USA)). The lyophilized samples were weighted and stored at −20 °C [31]. Deionized water was used to rehydrate to the desired concentration prior to use.

### 2.5. Antagonistic Activity Against S. aureus

The antagonistic activity tests of *L. brevis* DPL5 against 30 pathogenic *S. aureus* strains were evaluated using an agar plug diffusion test [36,37]. Briefly, 24 h liquid culture of *L. brevis* DPL5 was streaked on two MRS agar plates and incubated anaerobically at 37 °C for 24 and 48 h in 2.5 L anaerobic jar with Anaerocult A reagent (Merck, Darmstadt, Germany). Overnight cultures of the tested *S. aureus* strains were diluted to 1.5 × 10^8^ CFU mL^−1^ and uniformly spread using a sterile cotton swab on Mueller Hinton Agar plates (Merck, Darmstadt, Germany). The plates were dried for 30 min at room temperature. Then, 0.25 cm^2^ agar plugs of 24 h and 48 h *L. brevis* cultures were cut out and placed on the previously streaked *S. aureus* plates. Following 24 h incubation at 37 °C, the antagonism was determined by the presence/absence of inhibition zones around the agar plugs. The agar well diffusion method was used to determine the activity of *L. brevis* DPL5 cell-free supernatant [38]. *L. brevis* DPL5 was cultivated in 100 mL MRS broth anaerobically for 24 h and 48 h, respectively. The cultures were centrifuged, and the supernatants were filtered through a syringe filter with a pore size of 0.22 µm. As described previously, a 50 µL cell-free supernatant was added into separate, 6 mm diameter holes in the agar gel of plates inoculated with *S. aureus*.

### 2.6. Minimum Inhibitory Concentration (MIC) Assay

MIC assays with LCFS were performed by microdilution in 96-well plates (Costar^®^, Corning, NY, USA) [39]. A serial dilution was performed starting with 80 μg∙mL^−1^ of the supernatant on Tryptic Soy Broth, supplemented with 1% (*w*/*v*) glucose, 2% (*w*/*v*) NaCl, and 3% human plasma (TSBGSP), containing 5 × 10^5^ CFU∙mL^−1^ of *S. aureus* per well [31]. The results were compared against a lyophilized medium (MRS) without *L. brevis* DPL5. TSBGSP without inoculum was used as negative (sterility) control, and TSBGSP containing 5 × 10^5^ CFU∙mL^−1^ of *S. aureus* was used as positive control. Each *S. aureus* strain was tested in triplicates. The microplates were incubated for 24 h at 37 °C. The optical density of each well was determined in a spectrophotometer Multiskan FC (Thermo Scientific, Shanghai, China) at 620 nm. The absorbance was corrected against the negative control. MIC was determined as the lowest concentration, which inhibits the growth of *S. aureus*. Then, 5 μL from the MIC well and the wells with the next two higher concentrations were spotted on a new sterile MH agar plate and incubated for 24 h at 37 °C to classify the inhibitory concentrations as bactericidal or bacteriostatic.

### 2.7. Microtiter Plate Assay and Anti-Biofilm Activity

The quantitative test for biofilm formation was conducted using tissue culture polystyrene 96-well microtiter plates with flat bottoms (Costar^®^, Corning, NY, USA), following the methodology outlined by Čuvalová and Kmet [25] with minor changes. Staphylococci were cultured on MH agar (Merck), and single colonies were subsequently transferred to TSBGSP to achieve a concentration of 1.5 × 10^8^ CFU.ml^−1^. The effect of LCFS on staphylococcal biofilm formation was tested by the addition of 100 μL of different concentrations of supernatant to the polystyrene microtiter plates. Eight serial dilutions were prepared, yielding concentrations of 160, 80, 40, 20, 10, 5, 2.5 and 1.25 mg∙mL^−1^. The cell density of *S. aureus* strains was adjusted to 1 × 10^6^ CFU.ml^−1^, and 100 μL was dispensed into the preloaded with 100 μL LCFS microplate wells to a final concentration of 0.5 × 10^5^ CFU.ml^−^ and a final volume of 200 μL. The plates were incubated statically for 24 h at 37 °C. After incubation, the plates were washed thrice with 250 μL of saline solution (0.85% NaCl). Adherent cells were subjected to staining with a 0.1% crystal violet solution (Himedia, Dindhori, India) for 10 min. The surplus stain was removed by filling the wells with a saline solution (0.85% NaCl). The adherent dye was solubilized with 70% ethanol. The optical density of the wells was assessed at 610 nm utilizing a Multiskan FC (Thermo Scientific, Shanghai, China). Control wells contained 100 μL culture medium and 100 μL of the tested strain without adding CFLS. After 24 h of incubation at 37 °C, biofilm was quantified in the same manner as described above. As described by Saidi and collaborators [30], the Optical Density (OD) of the negative control well was recorded as ODc, and the OD of tested wells as ODt. The positive control tests with *S. aureus* demonstrated that all 30 strains had an ODt/ODc ratio > 4, classifying them as strong biofilm producers.

### 2.8. DNA Extraction, Sequencing, and Assembly

DNA was extracted from isolate DPL5 using the QIAamp DNA Microbiome Kit (QIAGEN, Hilden, Germany). The concentration and quality of the extracted DNA were evaluated using a Qubit 4 fluorometer (Thermo Fisher Scientific, Waltham, MA, USA) and agarose gel electrophoresis. To prepare long-read Oxford Nanopore Technology (ONT) libraries, the Ligation Sequencing Kit SQK-RBK114 (Oxford Nanopore Technologies, Oxford, UK) was utilized with 200 ng of total DNA following the manufacturer’s protocol. Subsequently, the prepared library underwent sequencing on a MinION device using an R10 flow cell (Oxford Nanopore Technologies, Oxford, UK). Base calling and quality checks were performed offline via Guppy v6.5.7 (Oxford Nanopore Technologies, Oxford, UK). Adapter sequences were removed using Porechop v0.2.4 with default settings (https://github.com/rrwick/Porechop, accessed on 10 January 2025).

De novo assembly was executed with Flye v2.9.2 under default settings, excluding reads shorter than 1000 bp. Polishing of the assembly was completed using Racon v1.4.21 (https://github.com/isovic/racon, accessed on 10 January 2025) and Medaka v1.8.1 (https://github.com/nanoporetech/medaka, accessed on 10 January 2025). The quality of the final assembly was verified using CheckM v1.1.6. A circular genome map of the single circular chromosome contig was visualized using the Proksee software (https://proksee.ca/, accessed on 10 January 2025).

#### 2.8.1. Genome-Based Identification and MultiLocus Sequence Typing (MLST)

The bacterial species identification of isolate DPL5 was achieved by calculating its average nucleotide identity (ANI) using FastANI. Additionally, the genome was analyzed for MLST using PubMLST (https://pubmlst.org, accessed on 10 January 2025). The Type (Strain) Genome Server (TYGS) was also utilized to construct a whole-genome sequence-based phylogenetic tree (https://tygs.dsmz.de/, accessed on 10 January 2025).

#### 2.8.2. Genome Annotation

The assembled genome of the DPL5 strain was submitted to NCBI Genomes for initial annotation through the Prokaryotic Genome Annotation Pipeline (PGAP) [40], which also assigned an accession number. The GenBank file generated by PGAP was further annotated using the Rapid Annotations using Subsystems Technology (RAST) [41] web server. Functional annotations were enhanced using the KEGG database and the BlastKOALA [42] tool by analyzing the predicted protein sequences from the PGAP GenBank file. The manual curation of genes related to probiotic properties was conducted using both RAST and KEGG-derived annotations. Carbohydrate-active enzymes (CAZymes) in the PU3 genome were identified using the cbCAN3 tool and the CAZy database (https://bcb.unl.edu/dbCAN2/, accessed on 10 January 2025). The genome of DPL5 was screened for antimicrobial resistance (AMR) and virulence factor (VF) genes using the Abricate tool (https://github.com/tseemann/abricate, accessed on 10 January 2025) with default settings, referencing the Comprehensive Antibiotic Resistance Database (CARD) [43], MEGARes DB [44], and the Virulence Factor of Bacterial Pathogens database (VFDB) [45]. Bacteriocin-related regions were predicted using the AntiSMASH 7.0 tool (https://antismash.secondarymetabolites.org/, accessed on 10 January 2025).

### 2.9. Statistical Analysis

Analysis of variance, *t*-test, graphical representation, and other results analyses were performed using Statistica 12 (StatSoft). The data for *L. brevis* are represented as mean ± standard deviation (SD). The data for biofilm formation in this study are expressed as mean ± 95% confidence intervals (*p* < 0.05) from the results of the 30 tested *S. aureus* strains. Probit analysis was performed using SPSS version 23 to estimate the MIC and minimal biofilm inhibitory concentration probability. All measurements were carried out in triplicate.

## 3. Results

### 3.1. Assimilation of Different Types of Carbon Sources and Osmotic Sensitivity

*Lactobacilli* have intricate nutritional needs for fermentable carbohydrates and derive energy through homofermentative or heterofermentative carbohydrate fermentation [46]. Their wide ecological distribution and adaptability to diverse habitats highlight their metabolic versatility, enabling the utilization of a broad range of carbohydrates [47]. Carbon catabolite repression (CCR) plays a key role in the competitive fitness of lactobacilli in natural environments like the gastrointestinal tract (GIT), where selecting the preferred carbon source significantly influences the growth rate and competition with other microorganisms [48]. We followed the uptake of different sugars in the *Lactobacillus brevis* DPL5 strain we studied. The results are presented in Figure 1. The analysis used the Biolog system and specific plates containing the studied sugars. We found good optical density in the medium’s presence of sugars such as lactose, sucrose, trehalose, and fucose. The strain has difficulty digesting sugars such as inosine, rhamnose, N-acetyl-D-mannosamine, N-acetyl-neuraminic acid, N-acetyl-D-galactosamine, and others.

*L. brevis* is frequently found in plant matter, but it has also been isolated from various other environments, such as beverages and the intestinal tracts of animals. This species is obligately heterofermentative, employs the phosphoketolase pathway, and possesses inducible glycolytic enzymes [49]. *L. brevis* has been shown to transport glucose, lactose, xylose, and galactose via proton symport systems. These proton symport systems are regulated through HPr, in which the transport mechanism is reversibly switched between proton symport and facilitated diffusion [50].

During industrial processes such as fermentation and freeze drying, lactic acid bacteria (LAB) often face significant osmotic stress [51]. Lactic acid bacteria primarily adapt to increased osmotic pressure through a two-phase response involving the accumulation of potassium ions (K^+^) and their counterions, such as proline and glycine betaine [52]. According to Sleator and Hill [53], the primary role of K+ accumulation in Gram-positive bacteria is to act as a signal, triggering the buildup of intracellular amino acids. We investigated the effect of different concentrations of sodium chloride −1%, 4%, and 9% on the growth of the *Lactobacillus* strain (Figure 2). No significant differences were established for the samples cultivated in 1% and 4% NaCl (*p* > 0.05). However, the highest concentration inhibited cell density for *L. brevis* DPL5 (*p* < 0.05).

### 3.2. Bile Salt Tolerance

The Food and Agriculture Organization and the World Health Organization define probiotics as “live microorganisms that, when consumed in sufficient quantities, provide health benefits to the host.” For probiotics to influence the intestinal environment, their population must reach a minimum threshold of 10^6^ to 10^8^ CFU per gram of intestinal content [54]. To establish themselves in the intestinal tract, these microorganisms must first withstand the harsh conditions of the digestive system, such as the acidic pH of the stomach and the presence of bile in the intestines. This ability to survive and thrive in such environments is a key characteristic of probiotics. In this regard, we investigated the influence of different percentage concentrations of bile salts added to the medium and followed the growth of the test strain *L. brevis DPL5—*Table 1.

### 3.3. Acid Tolerance

LAB employ various strategies to tolerate acidic conditions. These include producing alkaline substances via the arginine dihydrolase system to counteract acidity, neutralizing protons with carbon dioxide generated through malolactic fermentation, and expelling protons using proton pumps such as F1-F0-ATPase [55].

Figure 3 illustrates the cell viability of *L. brevis* DPL5 under various conditions. The viability of the cells at lower pH values is higher in the presence of low concentrations of pepsin in the medium compared to the viability that we reported in the absence of the enzyme. At pH around 5 and 7, no significant difference in survival is found in either variant.

Interestingly, pepsin demonstrated varying effects on cell viability. While pepsin is generally recognized for reducing microorganism viability through its proteolytic activity [56], exposure to pepsin increased the viability of strain DPL5 cells (Figure 3). This finding aligns with a previous study showing an increased viability of *Bifidobacterium animalis* subspecies upon exposure to pepsin. Although the precise mechanisms by which pepsin enhances the acid tolerance of lactic acid bacteria remain unclear, earlier research suggested that pepsin might aid in maintaining pH balance by supporting the activity of H^+^-ATPase in *Bifidobacterium animalis* subsp. lactis. This is thought to involve pepsin enhancing the proton pump’s function through ATP production [57]. While this hypothesis is yet to be validated, our findings support a similar explanation. Surviving the highly acidic environment of the gastrointestinal tract (GIT) is the first hurdle probiotic strains face, as gastric acid in the human stomach maintains a pH of approximately 1.5–3 [58]. To endure these extreme conditions in the gut, probiotic strains must possess acid tolerance. Many probiotics, particularly lactic acid bacteria (LAB), exhibit this tolerance and are essential in the fermentation industry [59]. Fermented food products are typically produced under specific acidic conditions facilitated by probiotic bacteria or yeast [60]. LAB such as *Lactobacillus brevis*, *Lactobacillus fermentum*, and *Leuconostoc mesenteroides* are used in fermenting durian fruit [61].

### 3.4. Determination of Antibiotic Susceptibility

The antibiotic susceptibility of *L. brevis* DPL 5 was assessed using the test paper agar diffusion method, and the findings are presented in Table 2. The strain demonstrated sensitivity to antibiotics such as tetracyclines, cephalosporins, and β-lactamase inhibitor complexes while showing resistance to Amikacin and Ceftriaxon. Aminoglycoside antibiotics inhibit protein synthesis and are known for their stability and broad-spectrum activity, primarily targeting Gram-negative bacteria while being less effective against Gram-positive bacteria. Experimental results for *L. brevis* DPL5 aligned with the previous literature [62].

The established resistance to ceftriaxone raises some concerns about the possibility for *L. brevis* DPL5 to act as reservoirs of transferable antibiotic resistance genes, but such resistance is not untypical [63]. Reports in recent years demonstrate that strains belonging to the *Lactobacillus brevis* group could often display high MIC values against β-lactams [64], and a wide occurrence of resistance to fourth and third-generation cephalosporins has been found even for other commercially available probiotic strains [65,66]. Moreover, the antibiotic resistance of lactobacilli can be considered ambivalently, as such resistance favors their survival during antibiotic therapy [67].

### 3.5. Antagonistic Effect of L. brevis DPL5 Using Plug-Diffusion and Well-Diffusion Methods

A preliminary antibacterial test was conducted to minimize time expenses. Due to the significant differences in the optimal culture conditions, the standard spot-on-the-lawn assay was ineffective for testing the possible inhibition effects of *L. brevis* DPL5 on *Staphylococcus aureus*. Instead, plug diffusion was used to test the antagonism [68]. Figure 4A indicates that the anaerobically cultured DPL5 for 24 h successfully combated *S. aureus*. The effects were probably mediated by direct cell competitive exclusion, as stated by Sikorska and Smoragiewicz [69]. The diameter of the inhibition zone (IZ) depends on the cultivation time. Increasing the cultivation time shows a positive effect with the highest IZ observed when 48 h culture DPL5 was applied (Figure 4B). Our findings paralleled the literature where [70] lactobacilli reach the maximum level of antimicrobial compound production during the stationary phase of their growth curve. Additional cultivation for up 72 h leads to a decrease in the antimicrobial activity (Figure 4C).

An additional well-diffusion assay with cell-free supernatant was performed to test the bioactivity of the extracellular compounds. Contrary to other studies [71], it shows that DPL5 possesses a better supernatant activity than cell-on-cell activity. The CFS significantly inhibited the growth of *S. aureus*. Results showed that the inhibition zones were larger when LCFS at 48 h of cultivation was used (21 mm) rather than at 24 h (11 mm) (Figure 4D,E). No significant differences in the diameter of IZ between MRSA and MSSA strains (*p* = 0.814) were found. The findings agree with several previous studies demonstrating the antimicrobial activity of lactobacilli [72] and their LCFS [73,74,75].

### 3.6. Antimicrobial Activity of DPL5

Antimicrobial activity is crucial for selecting probiotic bacteria as natural antagonists against pathogenic microorganisms [23]. Antimicrobial susceptibility tests revealed that the lyophilized CFS of *L. brevis* DPL5 had a strain-specific inhibitory effect on *S. aureus* cell growth. The MIC was in the range of 10 mg·mL^−1^ to 20 mg·mL^−1^ with an estimated probability (IBM SPSS Software) for MIC_99_ of 29.9 mg·mL^−1^ (Table 3). However, MIC had mostly a bacteriostatic (BS) effect on the tested nasal strains. The bactericidal concertation (BC) was dose dependent. For only 20% of the strains, BC was equivalent to MIC, while for the majority of strains, it reached up to 80 mg·mL^−1^. One-way ANOVA again showed no significant differences between MRSA and MSSA strains (*p* > 0.05), so subsequent analyses consider the two groups together. Based on the PROBIT analysis, the BC_99_ was calculated as 69 mg·mL^−1^. *L. brevis*, as a member of the genus Lacobacillus, is known for its potential to inhibit *S. aureus* [76,77], along with other species, such as *L. fermentum*, *L. plantarum*, *L. acidophilus*, *L. casei L. rhamnosus* and others [31,75,78,79]. In most cases, LAB are found to compete for colonization of the epithelial cells and control the overgrowth of pathogens [80], and only a few of them study the effect of extracellular extracts. Such antimicrobial activity for *L. brevis* is often a result of the production of organic acids (lactic and acetic acid) and/or bacteriocins, which inhibit the growth of Gram-positive pathogens [23,81]. Therefore, the demonstrated bactericidal properties of the LCFS of DPL5 highlight its probiotic potential.

The biofilm formation of *S. aureus* is considered a serious problem for public health and causes an increase in antibiotic resistance in chronic diseases and wound infection [82]. Currently, probiotic LAB metabolites are being evaluated along with natural compounds as an alternative to chemical agents to eradicate biofilm [22,83]. We investigated whether the LCFS from DPL5 facilitates the eradication of *S. aureus* biofilms. When strains were simultaneously treated with LCFS at 37 °C for 48 h, a concentration of 20 mg·mL^−1^ or higher led to a complete reduction in biofilm formation (*p* < 0.0001). The results are presented in Figure 5. The effect of lower concentrations was dose dependent, where 10 mg·mL^−1^ fully suppressed the biofilm formation in 32% of the strains (*p* < 0.0001) and significantly reduced the OD in the other 68% (*p* < 0.001). Concentrations of the LCFS as low as 5 mg·mL^−1^ significantly decreased the strength of the biofilms (*p* < 0.05) and were able to turn strong biofilms into weak ones. These results suggest that DPL5 LCFS effectively interferes with the biofilm formation of *S. aureus*. The estimated probability values for biofilm inhibitory concentration (BIC_99_) are presented in Table 2. These results conform to those of previous studies conducted on other lactobacilli [31,71,79]. In the present research, the established activity of LCFS subjected to pH neutralization showed that the antimicrobial activity is not a result of the production of organic acids. It suggests that the studied *L. brevis* DPL5 probably produced active metabolites against *S. aureus*. Moreover, the measurements of cell density that were obtained in conjunction with the examination of biofilm formation demonstrated that lower concentrations, which were less than the BIC_50_ threshold, did not result in the death of the nasal strains of *S. aureus*. According to Melo and collaborators [31], this demonstrates that the death of the pathogen did not decrease biofilm development. Such a statement is supported by the findings that the CFS from lactobacilli significantly down-regulates the virulence genes in *S. aureus* and interferes in the process of cell-to-cell communication [75,84], which leads to the inhibition of biofilm formation even without causing cell death. Furthermore, it confirms the production of modulatory chemicals capable of interfering with the pathogen’s ability to create biofilms.

### 3.7. Genomic Exploration of L. brevis DPL5 through Whole Genome-Sequencing

The genome of *L. brevis* DPL5 comprises a single circular chromosome measuring 2,334,043 base pairs (bps) with a coverage of ×10^5^ and a guanine–cytosine (GC) content of 46.26%. Additionally, the strain contains two mobilizable plasmids with sizes of 44,900 bp and 30,428 bp and GC content 41.10% and 45.88%, respectively. The complete genomic sequences have been deposited in the NCBI database under accession numbers CP144904 for the chromosome and CP144903 and CP144905 for the plasmids.

Multilocus Sequence Typing (MLST) is a widely used technique for identifying and typing LAB [85]. This method involves analyzing the sequences of multiple housekeeping genes to generate a distinct allelic profile for each microorganism. MLST is highly regarded for its accuracy and for producing results that can be easily compared and shared across different studies [86]. For the DPL5 genome, the analysis through PubMLST confirmed with 100% certainty that it belongs to the *L. brevis* species. Additionally, phylogenomic analysis via genome-to-genome comparisons in TYGS revealed that the DPL5 strain clusters with other representative *L. brevis* strains in the database (Figure 6). These findings further confirm the classification of DPL5 as a member of the *L. brevis* species.

The strain DPL5 exhibits a robust genomic profile that underscores its potential as a versatile and resilient probiotic candidate. Notably, the strain is equipped with 50 genes dedicated to stress response, highlighting its adaptability and potential to survive in harsh and fluctuating environments such as the gastrointestinal tract [87]. This arsenal of stress-related genes likely enhances its resilience against oxidative stress, osmotic pressure, and other hostile conditions, ensuring its viability and functionality in probiotic applications. Detailed analysis showed that the genome harbored genes with probiotic activity for acid and bile tolerance, temperature stress, adhesion, antioxidant activity, and immunomodulation [88] (Table 4).

Equally impressive is the presence of 112 genes associated with the metabolism of cofactors, vitamins, prosthetic groups, and pigments. These genes suggest that DPL5 can synthesize or metabolize a variety of essential micronutrients, contributing not only to its survival but also potentially to the nutritional enhancement of its host [89]. This capability could be particularly beneficial in promoting gut health, aiding in the recovery of vitamin deficiencies, and supporting the overall metabolic balance of the host organism.

The DPL5 strain’s extensive genetic repertoire, comprising 251 carbohydrate and 272 protein metabolism genes, underscores its ability to utilize diverse dietary substrates and efficiently support amino acid turnover. The 103 genes involved in the cell wall and capsule biosynthesis highlight structural robustness and possible immune-modulatory properties [90]. Furthermore, the strain features 85 genes for DNA metabolism and 84 for RNA metabolism, which is indicative of efficient genomic maintenance and transcriptional regulation as well as critical for adapting to environmental shifts. A detailed genome annotation table can be found in Appendix A.

The genome of the *L. brevis* strain DPL5 reveals the presence of a few antimicrobial resistance (AMR) genes, including those encoding beta-lactamase classes A and C. Additionally, most Lactobacillus species exhibit intrinsic resistance to aminoglycosides [91], which is a characteristic corroborated by our antimicrobial susceptibility analysis and the genome identification of AMR genes such as ykkC, ykkD, and aacC. Furthermore, a comprehensive genomic scan using the ABricate tool and the Virulence Factor Database (VFDB) revealed no virulence factors. The absence of virulence genes in the DPL5 strain is a critical factor supporting its safety and potential as a probiotic. Virulence genes are often associated with pathogenicity, posing risks to host health, especially in immunocompromised individuals [92]. The lack of such genes ensures that DPL5 does not contribute to infections or disrupt the host’s natural microbiota. This safety profile is essential for probiotics, which promote health without adverse effects. Therefore, the absence of virulence genes in DPL5 underscores its suitability for applications in food, health supplements, and therapeutic interventions.

CAZy analysis is crucial for new bacterial strains, providing insights into their metabolic potential and ecological roles. By identifying specific carbohydrate-active enzymes, researchers can predict the strain’s ability to degrade, utilize, or transform various polysaccharides, which is essential for understanding its functionality in different environments. This information is particularly valuable in designing probiotics or industrial applications where targeted carbohydrate metabolism is critical. Furthermore, CAZy profiling can help identify unique or novel enzymes, paving the way for biotechnological innovations in food, biofuel, and pharmaceutical industries [93].

The analysis of CAZy enzymes in the DPL5 strain reveals a diverse repertoire of carbohydrate-active enzymes, highlighting its potential for utilizing a wide range of carbohydrates. The strain exhibits enzymatic capabilities for breaking down complex polysaccharides, including xylan, starch, and chitin, suggesting its adaptability to plant- and animal-derived substrates [94]. It also shows activity related to sucrose metabolism, indicating its ability to process common dietary sugars. The presence of enzymes targeting beta-glucuronan, beta-glucan, and beta-galactan further underscores its versatility in degrading various beta-linked polysaccharides. Additionally, enzymes associated with the metabolism of exo-polysaccharides, alpha-glucan, arabinan, and alpha-galactan expand the strain’s carbohydrate-processing spectrum. These features enhance the strain’s resilience in diverse environments and suggest its potential applications in probiotics and prebiotic development, where efficient carbohydrate utilization is a key trait.

The AntiSMASH tool can identify a wide range of biosynthetic gene clusters (BGCs), including those responsible for producing bacteriocins, which are ribosomally synthesized antimicrobial peptides produced by bacteria. The results of the DPL5 strain showed lanthipeptide class IV, the T3PKS region and the RiPP-like region, all of which are chromosome encoded (Figure 7).

Lanthipeptides are a class of ribosomally synthesized and post-translationally modified peptides (RiPPs) known for their antimicrobial properties. Class IV lanthipeptides are characterized by their unique structural features, including multiple lanthionine rings. Identifying this class in a strain suggests the ability to produce compounds that can inhibit the growth of competing microorganisms, which may confer a competitive advantage in various ecological niches. RiPPs encompass a broad category of natural products derived from ribosomal synthesis followed by enzymatic modifications. The detection of RiPP-like regions suggests the presence of additional pathways for producing bioactive peptides that could function as antimicrobials or signaling molecules [95,96]. These compounds often play critical roles in microbial interactions and defense mechanisms. The diversity of RiPP-like regions is notable with different LAB genera exhibiting varying capacities for producing these compounds. The analysis showed that 28 out of 55 genera contained at least one RiPP per genome, indicating a widespread ability to synthesize these bioactive peptides. Furthermore, the study revealed that a high percentage of these gene clusters were specific to certain genera or species, suggesting that niche adaptation plays a critical role in distributing these biosynthetic capabilities. For instance, 92.6% of the identified gene clusters were genus-specific, underscoring the potential for unique metabolite production tailored to specific ecological niches [97]. The presence of RiPP-like regions in LAB is particularly significant for probiotic applications. These compounds not only possess antimicrobial properties but also play roles in shaping the microbiota by inhibiting pathogens and supporting beneficial microbial populations.

Type III polyketide synthases (T3PKS) are versatile enzymes involved in the biosynthesis of polyketides, which are secondary metabolites with diverse biological activities, including antimicrobial, antifungal, and anticancer properties. The presence of T3PKS regions indicates that the strain can synthesize complex polyketide structures that may have significant pharmaceutical potential [97].

## 4. Conclusions

The observations of the presented in vitro study indicated that the *L. brevis* DPL5 strain can survive in the gastrointestinal tract and industrial processes such as fermentation and freeze-drying processes. The strain demonstrated a significant antagonistic effect against nasal strains of *S. aureus*. The lyophilized cell-free supernatant of this LAB markedly reduced biofilm formation at concentrations <BIC50 and exhibited pronounced antimicrobial action at elevated concentrations (20 mg·mL^−1^), leading to the complete eradication of biofilms. The results suggest the production of bioactive extracellular compounds that are capable of interfering with the pathogen’s ability to create biofilms. The study highlights the probiotic potential of *L. brevis* DPL5. Conducting further studies on the extracellular production of biologically active metabolites by the strain studied would create an opportunity to develop biological alternatives to chemical dosage forms for topical treatment and the prevention/treatment of infections caused by biofilm-forming strains of *S. aureus*. In addition, genome-wide sequencing and in silico analysis of the *L. brevis* DPL5 using Nanopore technology reveals that it contains over 50 genes associated with probiotic activity, which may play a crucial role in its survival within the gut environment. The genome also includes multiple biosynthetic gene clusters that exhibit antibacterial properties.

## Figures and Tables

**Figure 1 microorganisms-13-00160-f001:**
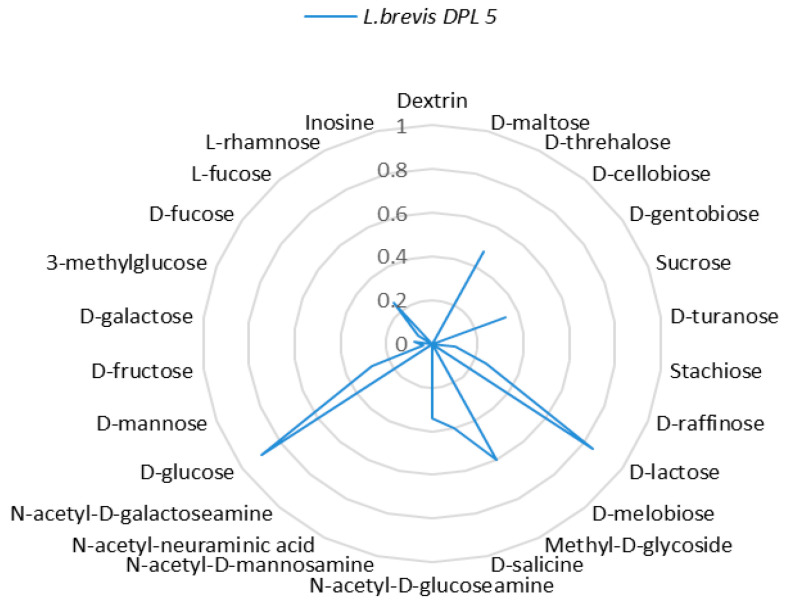
Optical density (600 nm) after culturing in the presence of different sugars in a *Lactobacillus brevis* DPL5.

**Figure 2 microorganisms-13-00160-f002:**
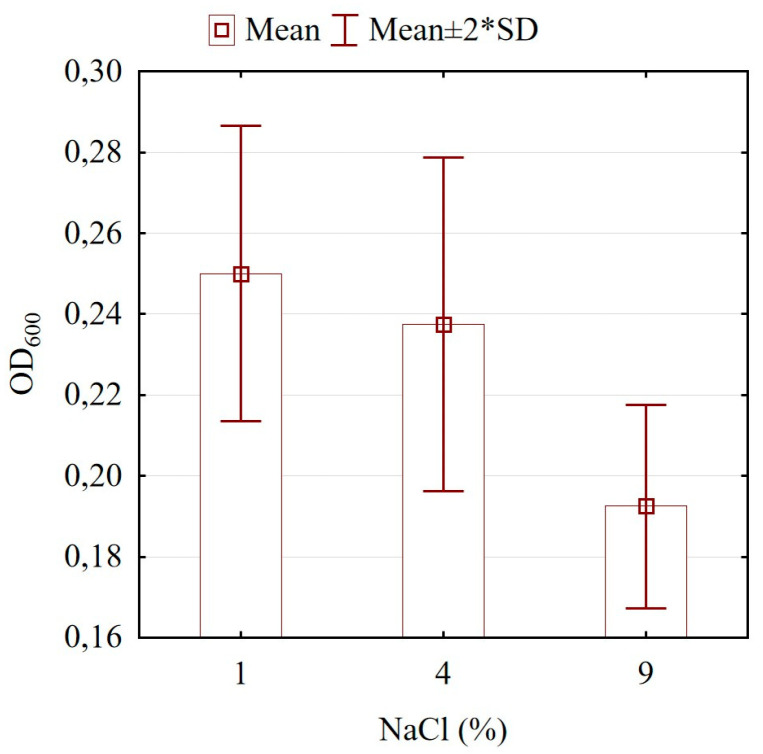
Effect of different concentrations of sodium chloride on the growth of *Levilactobacillus brevis* DPL5.

**Figure 3 microorganisms-13-00160-f003:**
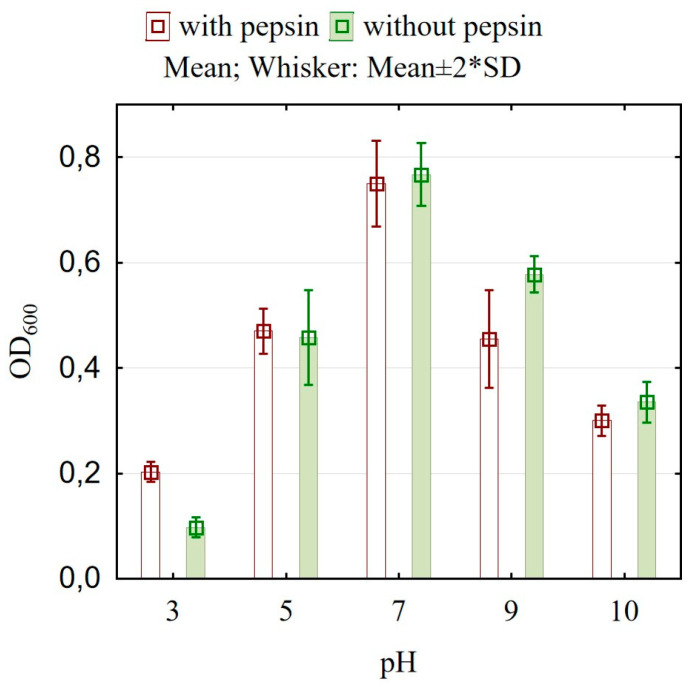
Survival of *L. brevis* DPL5 in different acidic conditions.

**Figure 4 microorganisms-13-00160-f004:**
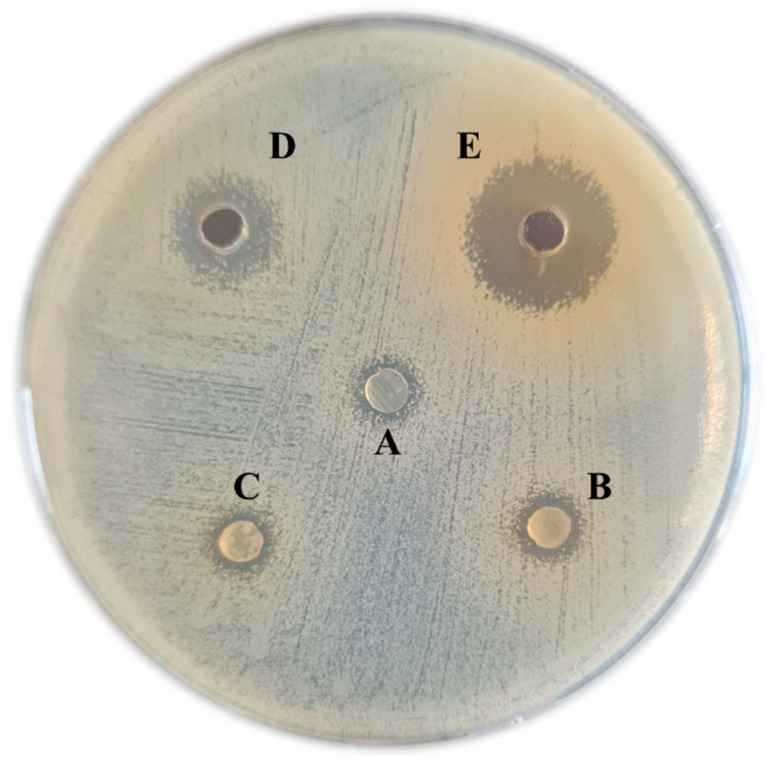
Antagonistic activity against nasal *Staphylococcus aureus* strain: Plug diffusion test with 24 h (**A**), 48 h (**B**), and 72 h (**C**) *L. brevis* DPL5 cultivated on MRS agar. Agar well diffusion test showing the activity of *L. brevis* DPL5 cell-free supernatant (50 µL) after anaerobic cultivation in MRS broth for 24 h (**D**) and 48 h (**E**).

**Figure 5 microorganisms-13-00160-f005:**
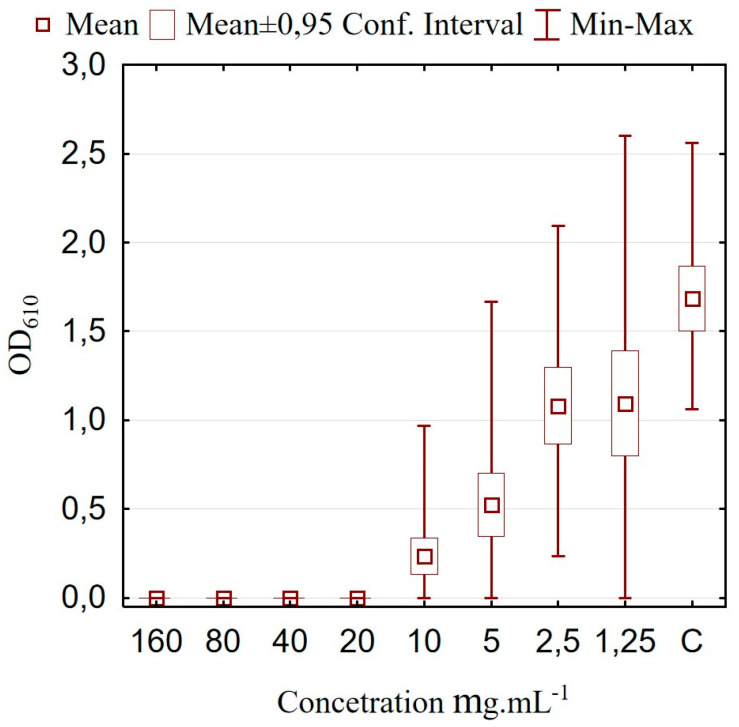
Effect of different concentrations of lyophilized *L. brevis* DPL5 cell-free supernatant on the intensity of biofilm formation in nasal *S. aureus* strains.

**Figure 6 microorganisms-13-00160-f006:**
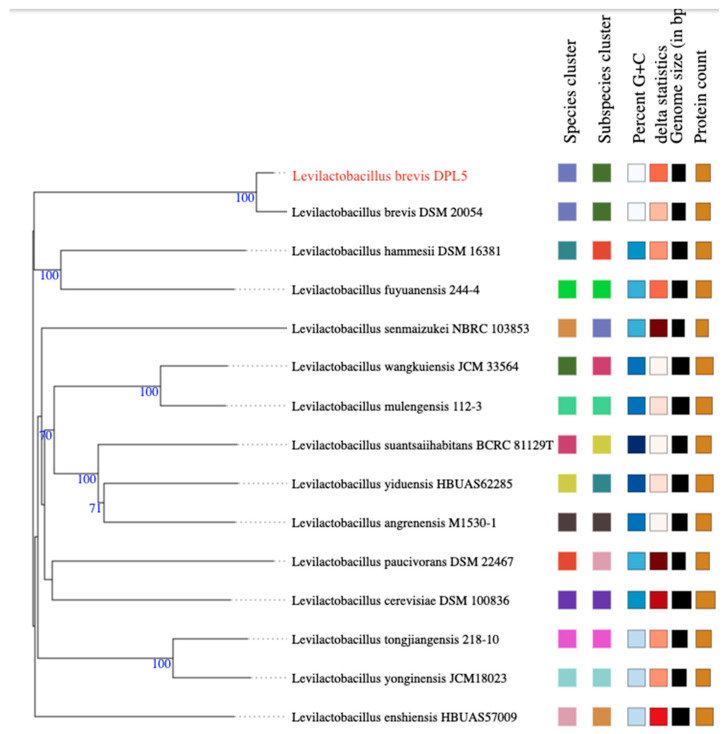
Confirmation of the *L. brevis* DPL5 via genome-to-genome comparisons in TYGS.

**Figure 7 microorganisms-13-00160-f007:**
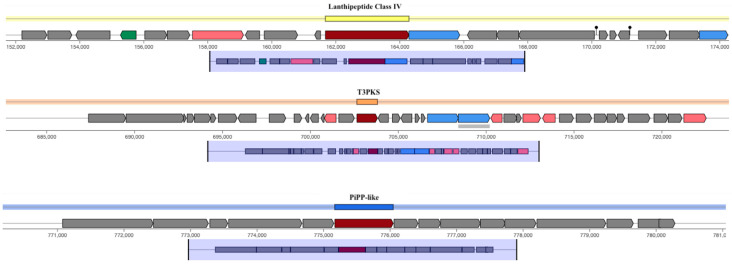
Genomic maps of the clusters of the lanthipeptide class IV, T3PKS region, and RiPP-like region of the *L. brevis* DPL5.

**Table 1 microorganisms-13-00160-t001:** Bile salt resistance of *L. brevis DPL5*, expressed as mean ± SD.

Medium	Bile Salts Concentration (%)		
	0.3%	1%	3%
MRS	72 ± 0.31	41 ± 0.23	6 ± 0.34
MRS with 7% lactose	84 ± 051	52 ± 0.24	10 ± 0.13

Note: Growth of strain DPL5 in MRS or MRS with lactose broth was determined by measuring the absorbance at a wavelength of 600 nm after incubation at 37 °C. Bile resistance is expressed as a percentage of growth of the control (bile-free growth). Results were obtained from duplicate samples and were representative of 2 independent trials. Mean values with unlike superscript letters within a line differed significantly (*p* < 0.05).

**Table 2 microorganisms-13-00160-t002:** The antimicrobial susceptibility results of *L. brevis* DPL5 expressed as mean ± SD.

Antibiotic Category	Name of Antibiotics	Dosage (µg)	Diameter of Inhibition Circle (mm)	Drug Sensitivity
Aminoglycosides	Amikacin	30	11.32 ± 0.41	R
Tetracyclines	Tetracycline	30	22.07 ± 1.15	S
Cephalosporins	Ceftazidime	10	19.03 ± 0.68	S
	Cefazolin	30	22.51 ± 0.98	S
	Ceftriaxon	30	9.61 ± 0.58	R
	Cefepime	30	21.45 ± 0.83	S
Quinolones	Ciprofloxacin	5	21.5 ± 1.25	S
β-Lactamase inhibitor complex	Ampicillin/Sulbactam	10	26.08 ± 1.05	S

Note: R indicates drug resistance; I indicates moderately sensitive; S indicates sensitive.

**Table 3 microorganisms-13-00160-t003:** Probability analysis for estimation of mean value and 95% confidence intervals for minimal biofilm inhibitory, minimal bacteriostatic, and bactericidal concentrations (mg·mL^−1^) for the 48 h cell-free *L. brevis* DPL5 lyophilized supernatant against nasal *S. aureus* strains based on the observed results.

	Biofilm Inhibitory Concertation			Bacteriostatic Concentration			Bactericidal Concentration		
	BIC_50_	BIC_75_	BIC_99_	MIC_50_	MIC_75_	MIC_99_	BC_50_	BC_75_	BC_99_
Estimate probability	9.847	13.589	29.908	26.23	31.64	44.88	28.304	36.656	69.048
Lower Bound	8.291	11.366	21.749	22.91	27.86	38.76	24.111	31.415	52.738
Upper Bound	11.831	17.704	53.759	30.45	37.42	55.71	33.229	46.198	117.907

**Table 4 microorganisms-13-00160-t004:** Probiotic gene markers identified in *L. brevis* DPL5.

Probiotic Activity	Genes
Acid and bile tolerance	*rpsS*, *pepF*, *kup*, *fabH*, *guaA*, *uvrA*, *dltA*, *dltC*, *dltD*, *pyk*, *recA*, *atpC*, *atpD*, *atpG*, *atpA*, *atpH*, *atpF*, *atpE*, *atpB*,
Acid stress/bile resistance/temperature	*grpE*, *dnaJ*, *luxS*, *dnaK*, *eno*, *pgk. pgi*, *tpiA*, *gap*, *uvrA*, *nhaC*
Adhesion	*tuf*, *lspA*
Antioxidant	*msrA*, *msrB*, *trxA*, *tpx*, *nrdH*, *mntH*
Bile resistance	*rplD*, *rpsC*, *rplE*, *rplF*, *rpsE*, *argS*, *lpdA*, *glnA*, *pyrG*
Cell wall formation or adhesion	*murA*
Cold stress	*rnr*
Immunomodulation	*dltB*, *dltC*, *dltD*
Ionic and heavy metal stress resistance	*corA*
Oxidative stress	*msrB*
Proteases	*clpX*
Temperature stress	*hrcA*, *hslV*, *htpX*

## Data Availability

Data are contained within the article and Appendix A.

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
