# Peer review of "Characterization and Probiotic Potential of Levilactobacillus brevis DPL5: A Novel Strain Isolated from Human Breast Milk with Antimicrobial Properties Against Biofilm-Forming Staphylococcus aureus"

_microorganisms, 2025, doi:10.3390/microorganisms13010160_

Round 1

Reviewer 1 Report

Comments and Suggestions for Authors

This article is well-written overall and thoroughly characterizes the probiotic potential of Levilactobacillus brevis DPL5. The methodology is sound, as it adequately assesses several microbiological tests that demonstrated probiotic traits and potential antibacterial and antibiofilm capabilities against S. aureus strains, as well as a genome analysis that gives interesting insights into this study. The study design is comprehensive, ensuring the reliability and relevance of the results. However, the main issue is regarding the selection of appropriate citations to support the study in parts of the introduction and discussion sections. Corrections could be made, and the authors should consider the following suggestions:

General:

Scientific names should be in italics. Please, revise thoroughly the manuscript.

Introduction

Ln35-49. This paragraph is well-written, and I suggest including experimental studies rather than review articles to support historical scientific evidence.

Ln75-81. I suggest including appropriate references to support this knowledge. 

Ln82-85. I suggest including appropriate references to support this knowledge.

M&M

Ln120. Additional information regarding S. aureus strains should be highly valuable because the abstract mentioned them as nasal isolates. Probably, a supplemental table containing such information could be included.

Ln. 181-182. What volume was added? 200 ul of culture plus a given volume of LCFS? Or 200 ul of culture containing a volume of LCFS? Please specify it.

Ln- 203-212. What final bacterial concentrations were used? Please specify it.

Results and Discussion 

Figure 3 could be improved. 

Ln376-377. It seems out of context, but it is essential, please provide a reference.

Ln473-542. This part of the discussion should include references to support explanations, hypotheses, and comparisons to accurately discuss.

Author Response

Comments 1: Scientific names should be in italics. Please, revise thoroughly the manuscript.

Response 1: We thank the reviewer for the recommendation. All scientific names are corrected and presented in italics

Comments 2: Ln35-49. This paragraph is well-written, and I suggest including experimental studies rather than review articles to support historical scientific evidence.

Response 2:We thank the reviewer for the valuable suggestion. We have included additional experimental studies to support the statements.

Comments 3: Ln75-81. I suggest including appropriate references to support this knowledge. 

Response 3: We have added a references in the text

Comments 4: Ln82-85. I suggest including appropriate references to support this knowledge.

Response 4: We have added a references in the text

Comments 5: Ln120. Additional information regarding S. aureus strains should be highly valuable because the abstract mentioned them as nasal isolates. Probably, a supplemental table containing such information could be included. -

Response 5: Thank you for the suggestion. We have added a supplementary table with information regarding the S. aureus strains.

Comments 6: Ln. 181-182. What volume was added? 200 ul of culture plus a given volume of LCFS? Or 200 ul of culture containing a volume of LCFS? Please specify it.

Response 6: Thank you for the question. We have improved the section.

Comments 7: Ln- 203-212. What final bacterial concentrations were used? Please specify it.

Response 7: Thank you for the question. We have corrected the paragraph to describe more clearly the procedure. 

Comments 8: Figure 3 could be improved. 

Response 8: Thank you for the suggestion. Figure 3 and 4 were corrected and standard deviation was also presented.

Comments 9: Ln376-377. It seems out of context, but it is essential, please provide a reference.

Response 9: We have added a reference in the text

Comments 10: Ln473-542. This part of the discussion should include references to support explanations, hypotheses, and comparisons to accurately discuss.

Response 10: Thank you for the valuable suggestions. We have improved the section. Furthermore,  some software tools do not have proper references/papers, and if that was the case the direct URL to the tool source is provided.

Reviewer 2 Report

Comments and Suggestions for Authors

In this study, a newly isolated Levilactobacillus brevis DPL5 strain from human breast milk. The aim of this study is to investigate the probiotic potential of L. brevis and to evaluate its antibacterial properties against S. aureus. The genomic structure of L. brevis was also determined. Results showed L. brevis DPL5 exhibited a powerful antagonistic effect against nasal strains of Staphylococcus aureus, significantly hindering biofilm formation. The content of this study is interesting and some modifications are recommended:

Abstract: The aim of this study is not clearly summarized. The authors have added too much discussion in the abstract, such as suggest, point out, etc. However, the key results of this study is not logically introduced. The last sentence is not suitable for abstract because the exploration of DPL5s bioactive metabolites” was not conducted in this study.

Methods: The methods have been introduced in detail. However, some sections lack proper references.

Results: The sequence of the results is not consistent with that of methods. Moreover, the data should be expressed as mean +/- standard derivation. Statistical analysis should be conducted to compare the results in the figures and tables.

The authors should avoid the expression of we... throughout the manuscript. When introduced the results, the authors have introduced too much background information. The background should be simply introduced, for example, lines 276-285, 304-314, etc. Instead, the results of this study is only simply described, which is inappropriate. For example, lines 286-292, 315-318, etc. So the writing of this manuscript must be improved.

Comments on the Quality of English Language

 The English could be improved to more clearly express the research.

Author Response

Comments 1: Abstract: The aim of this study is not clearly summarized. The authors have added too much discussion in the abstract, such as “suggest”, “point out”, etc. However, the key results of this study is not logically introduced. The last sentence is not suitable for abstract because “the exploration of DPL5’s bioactive metabolites” was not conducted in this study.

Response 1: Thank you for the valuable suggestions.The abstract was improved according to the reviewer suggestions.

Comments 2: Methods: The methods have been introduced in detail. However, some sections lack proper references.  

Response 2: Thank you for the valuable suggestions. We have added the relevant references. However, some software tools do not have proper references/papers, and if that was the case the direct URL to the tool source is provided.

Comments 3: Results: The sequence of the results is not consistent with that of methods.  

Response 3: Thank you for the suggestion. We have rearranged the text to ensure that the results correspond to the sequence of procedures.

Comments 4: Moreover, the data should be expressed as mean +/- standard derivation. Statistical analysis should be conducted to compare the results in the figures and tables.

Response 4: Thank you for the suggestion. Figure 3 and 4 were corrected and standard deviation was also presented, and also included SD in the Tables. The outcomes of a one-way ANOVA for the statistical assessment of the reported results are also included.

Comments 5: The authors should avoid the expression of “we...” throughout the manuscript. When introduced the results, the authors have introduced too much background information. The background should be simply introduced, for example, lines 276-285, 304-314, etc. Instead, the results of this study is only simply described, which is inappropriate. For example, lines 286-292, 315-318, etc. So the writing of this manuscript must be improved.

Response 5: Thank you for the valuable suggestions. We have improved the section.

Comments 6: The English could be improved to more clearly express the research.

Response  6: The manuscript language was improved.

Reviewer 3 Report

Comments and Suggestions for Authors

This study evaluates the antimicrobial activity of L. brevis-derived cell-free supernatant and its ability to prevent S. aureus biofilm formation. The manuscript is well-written, clear, and well conceptualized. However there some points that should be cleared before the manuscript is suitable for pubblication.

1)    The authors must provide further details concerning the strains of S. aureus used in the experimental procedures. In particular, it is necessary to ascertain whether the strains were meticillin-resistant.

2)    What explanations can be put forward by authors for L. brevis's resistance to third-generation cephalosporins?

3)    How do the authors explain that CFS has better inhibitor activity than cell-to-cell activity?

4)    It would be of interest to know whether the authors have also tested the antibacterial activity of LCFS or DPL5 on microrganisms other than S. aureus.

5)    How does the supernatant inhibit biofilm formation without killing the micro-organism? The author should provide a gel of the CSF.

6)    In view of the fact that the mechanisms by which extracellular compounds are capable of eradicating biofilms have yet to be investigated, it is recommended that the authors should at least describe the nature of the components present in the supernants (proteins, sugars, glycoproteins etc.).

Author Response

Comments 1: The authors must provide further details concerning the strains of S. aureus used in the experimental procedures. In particular, it is necessary to ascertain whether the strains were meticillin-resistant.

Response 1: Thank you for the suggestion. We have added a supplementary table with information regarding the S. aureus strains.

Comments 2: What explanations can be put forward by authors for L. brevis's resistance to third-generation cephalosporins?

Response 2: Thank you for the question. We have improved the section and added information concerning the antimicrobial resistance.

Comments 3:  How do the authors explain that CFS has better inhibitor activity than cell-to-cell activity?

Response 3: Thank you for the question. We believe that the differences found in the activity of the whole cell fraction and the cell-free supernatant are an indicator for low reliability of the plug-diffusion assay when working with microorganisms with different physiological requirements, such as L. brevis and S. aureus. Culturing under conditions optimal for S. aureus results in a loss of cellular activity in L. brevis and the observed bactericidal effect is due to components already synthesized in the medium. The well-diffusion assay allows working with a larger supernatant volume, leading to more pronounced results. Both assays are qualitative screening assays aimed at determining the possible potential of L. brevis for subsequent studies and are therefore not discussed in detail.

Comments 4:  It would be of interest to know whether the authors have also tested the antibacterial activity of LCFS or DPL5 on microrganisms other than S. aureus.

Response 4: Thank you for the valuable suggestions. We heve conducted preliminary studies with Gram-negative biofilm-forming species. However, the significant differences in quorum-sensing and biofilm-forming mechanisms between Gram-positive and Gram-negative species make results difficult to compare. S. aureus was selected as a species of significant scientific and medical interest.

Comments 5:  How does the supernatant inhibit biofilm formation without killing the micro-organism? The author should provide a gel of the CSF.

Response 5: We sincerely thank the reviewer for the question. The supernatant's action is dose-dependent, with low concentrations preventing biofilm formation and higher concentrations exhibiting a bactericidal effect on the examined bacteria. We incorporated supplementary references indicating that sublethal concentrations influence gene regulation by inhibiting the transcription of specific virulence components, including the AGR operon, which governs the quorum sensing (QS) system, so corroborating our findings. Our investigation has shown that additional elevations in LCFS concentration result in cellular death. The findings are displayed in Table 2. At this point, the whole cell-free supernatant is characterized by high protein concentration. The gel electrophoresis data have not been displayed due to the absence of identifiable bands. Nevertheless, additional research is required to validate this effect on gene regulation.

Comments 6:  In view of the fact that the mechanisms by which extracellular compounds are capable of eradicating biofilms have yet to be investigated, it is recommended that the authors should at least describe the nature of the components present in the supernants (proteins, sugars, glycoproteins etc.).

Response 6: We sincerely thank the reviewer for highlighting the importance of understanding the components present in the supernatants. The supernatants used in our study likely include a vast majority of the extracellular proteome as well as secondary metabolites, signaling molecules, etc. We acknowledge that a detailed exploration of these components is critical to fully elucidating their role in biofilm eradication. However, such an investigation requires extensive and time-intensive proteomic and metabolomic analyses, which are beyond the current scope of this work. The genomic exploration shows that DPL5 harbors several biosynthetic gene clusters with antimicrobial properties which may focus the future work in this direction.

Round 2

Reviewer 2 Report

Comments and Suggestions for Authors

 Accept in present form

Reviewer 3 Report

Comments and Suggestions for Authors

In the revised submission the authors have satisfactorily addressed my comments